# AMPLE: An R package for capacity building on fisheries harvest strategies

**Finlay Scott** *, **Nan Yao, Robert Dryden Scott**

SPC (Pacific Community), Nouméa, New Caledonia

* finlays@spc.int

## Abstract

Fisheries harvest strategies are formal frameworks that represent a best-practice approach for sustainable fisheries management. A key component of a harvest strategy is a 'pre-agreed rule', known as a harvest control rule (HCR), that sets fishing opportunities, e.g. catch limits, based on an estimate of fish stock status, e.g. estimated stock biomass. The harvest strategy development process is driven by stakeholders who are required to make a range of informed decisions, including on the selection of the preferred HCR. Capacity building may be required to facilitate the stakeholder engagement, particularly regarding the technical components of harvest strategies, including HCRs. The AMPLE package for R provides three interactive apps that support capacity building and stakeholder engagement on HCRs. These apps have been used during in-country national workshops around the western and central Pacific Ocean (WCPO) to support the development of harvest strategies for the Western and Central Pacific Fisheries Commission. These apps include several novel features: they take users from a gentle introduction to how HCRs work, to using methods for testing, comparing and selecting a preferred HCR from a suite of candidates. They include an introduction to the impact of uncertainty on the performance of an HCR, introduce performance indicators and discuss methods for selecting the preferred HCR based on management objectives. As such they provide a more detailed overview of HCRs than currently existing alternatives. These apps provide an effective platform for hands-on learning and have proven to be successful at supporting capacity building on HCRs in the WCPO. For example, using them for group activities and competitions stimulated productive discussions and increased understanding. As the model fishery in AMPLE is generic and not based on a real example, the apps will also be of interest to scientists, managers and stakeholders developing harvest strategies in other regions.

## Introduction

A harvest strategy is a formalised framework for fisheries management that provides the best chance of achieving management objectives, both for the fishery and the fish stock. It represents a best-practice approach for fisheries management [1] as well being an important consideration for Marine Stewardship Council certification [2].

Central to a harvest strategy is a management procedure that sets fishing opportunities, such as effort or catch limits, using an estimate of stock status, such as the current biomass [3].

**Data Availability Statement:** The software described in this manuscript does not use a data set. All data is generated internally by the software. The source code is available at Github: https://github.com/PacificCommunity/ofp-sam-amped/tree/master/AMPLE.

**Funding:** This was work was supported by the New Zealand Ministry of Foreign Affairs and Trade (MFAT) (https://www.mfat.govt.nz/) funded project "Pacific Tuna Management Strategy Evaluation". All authors (FS, NY, RDS) were supported by this project. The funders had no role in study design, data collection and analysis, decision to publish, or preparation of the manuscript.

**Competing interests:** The authors have declared that no competing interests exist.

A management procedure has three components: data collection, a stock status estimation method, and a pre-agreed rule known as a harvest control rule (HCR). All three components are agreed together. The use of a management procedure to set fishing opportunities removes the need for fisheries managers and stakeholders to routinely negotiate and agree the level of fishing. This streamlines the management process and reduces the risk of making decisions that focus only on short-term concerns.

Candidate management procedures are tested in a simulation framework, known as Management Strategy Evaluation (MSE), and their relative performance compared using performance indicators [4, 5]. This allows the selection and adoption of the preferred management procedure.

The development of a harvest strategy is driven by fishery managers and stakeholders who are required to make a range of informed decisions during the development process, including deciding which management procedure should be adopted. Capacity building may be required to facilitate the active participation of decision makers [6].

The tuna fishery in the western and central Pacific Ocean (WCPO) provides 52% of the global tuna catch and is the largest tuna fishery in the world [7]. The four main target species, skipjack, yellowfin, bigeye and albacore, are caught by a diverse range of fleets and fishing methods. For many countries in the WCPO, tuna fisheries are crucial for their economies, livelihoods and culture [8–11]. However, tuna stocks will only be a dependable and renewable resource if they are managed responsibly.

The Western and Central Pacific Fisheries Commission (WCPFC) is a regional fisheries management organisation that has oversight of tuna fisheries management in the WCPO region. WCPFC has 26 member countries and territories, 7 participating territories and 9 cooperating non-members. In 2014 the WCPFC agreed to a work plan for adopting a harvest strategy approach for the four main tuna species (CMM 2014–06).

Harvest strategies are new to the WCPO region and many WCPFC members are unfamiliar with their details. There is, therefore, a strong need for capacity building to facilitate stakeholder engagement. It is essential that all members understand HCRs and performance indicators so that they can actively participate in the management procedure selection process. This is challenging given the diversity and number of WCPFC members.

The WCPFC harvest strategy development work is being carried out by the Pacific Community (SPC) along with the Forum Fisheries Agency (FFA). Capacity building and stakeholder engagement has focused on the delivery of national workshops around the region, including in Kiribati, Fiji, Papua New Guinea, Solomon Islands, Palau and Tonga [12].

A range of methods may be required to communicate the technical aspects of harvest strategies, such as HCRs. The use of interactive apps to support capacity building can be effective, for example, the *ToyTuna MSE* tool, developed under the GEF/FAO ABNJ project [13, 14], as they provide a hands-on learning experience. Interactive tools are also available for performing MSE, such as the *DLMtool*, [15] which may be useful for capacity building.

Three new apps were developed for the workshops using R and the Shiny package [16, 17]:

- *Introduction to Harvest Control Rules*

- *Measuring performance*

- *Comparing performance*

It is essential that any apps used in the WCPO workshops are appropriate for WCPFC members. These apps therefore include several novel features that make them particularly suitable for participants with no pre-existing knowledge of HCRs. For example, they include a very gentle introduction to how HCRs work that allows users to go step-by-step through the

HCR process. There is also a basic introduction to the impact of uncertainty on the performance of an HCR that allows users to run individual stochastic simulations that, taken together, can be used to measure performance. As such these apps provide a more detailed overview of HCRs than currently existing alternatives.

The apps are not tools for developing the HCRs that will be adopted across the WCPO (they include only a simple biological model with standard HCRs), but tools for capacity building and communication so that stakeholders can actively participate in the harvest strategy development process.

During the workshops, these apps proved to be a popular and successful platform for learning. Using them for group activities and competitions helped bring workshops to life and stimulated constructive discussions. The AMPLE package collects these apps into a single R package so that they can be easily distributed and used by others. The package is distributed under a GPL (>= 3) licence.

The model fisheries in the apps are not based on a real-world fishery. They are therefore not only applicable to the WCPO but may also be of interest to stakeholders developing harvest strategies in other regions.

## AMPLE features

The three apps in AMPLE share the same underlying model fishery, implemented as a Schaefer surplus production model with an annual time step [18, 19]:

$$B_{y+1} = B_y + g\left(B_y\right) - C_y \tag{1}$$

where $g(B)$ is the surplus production given as $g(B) = r\,B(1 - B\,/\,K)$, $B_y$ is the biomass at the start of the year $y$, $C_y$ is the total catch during the year $y$, $r$ is the intrinsic growth rate of the stock and $K$ is the carrying capacity, or average biomass level prior to exploitation of the stock.

To make the apps generic and applicable to all fisheries, the presented catches have no units and the presented biomass is divided by $K$ to scale it between 0 and 1 so that it is relative to the maximum unfished biomass.

The life history of the stock can affect the performance of an HCR. To help explore this, users are able to select from three different life histories, defined by the value of $r$ (slow = 0.2, medium = 0.6 or fast = 1.0). $K$ is then scaled to give a Maximum Sustainable Yield (MSY) of 100, using the relationship $MSY = r\,K\,/\,4$.

The current status of the stock can also influence the choice of HCR, e.g. a stock rebuilding plan may be required. To support this, users can select from three catch histories for the fishery: under- (catch is constant at 2/3 MSY), fully- (constant at MSY), or over-exploited (increasing from 3/4 to 4/3 MSY). Normally distributed random noise with a standard deviation of 0.1 is applied to the historical catches to provide some variability.

The life history and catch history settings, as well as the duration of the projections, can be changed under the **Settings** tab of each app.

In each year, the fishing opportunities for the following year are managed through an HCR that sets an annual catch limit based on the estimated stock biomass. Currently, a single type of HCR is available (called a 'threshold-catch' in the app, sometimes known as a hockey stick), the shape of which is determined by four parameters: *Cmax* (the maximum allowable catch), *Cmin* (the minimum allowable catch), *Blim* (the biomass below which the minimum catch is set) and *Belbow* (the biomass above which the maximum catch is set) (Fig 1). In the real world, *Blim* would likely correspond with a biological limit reference point (such as the Limit Reference Point, see below), below which the stock is considered to be at risk. It may impossible or impractical to remove all fishing pressure if the stock falls below this level of biomass, therefore

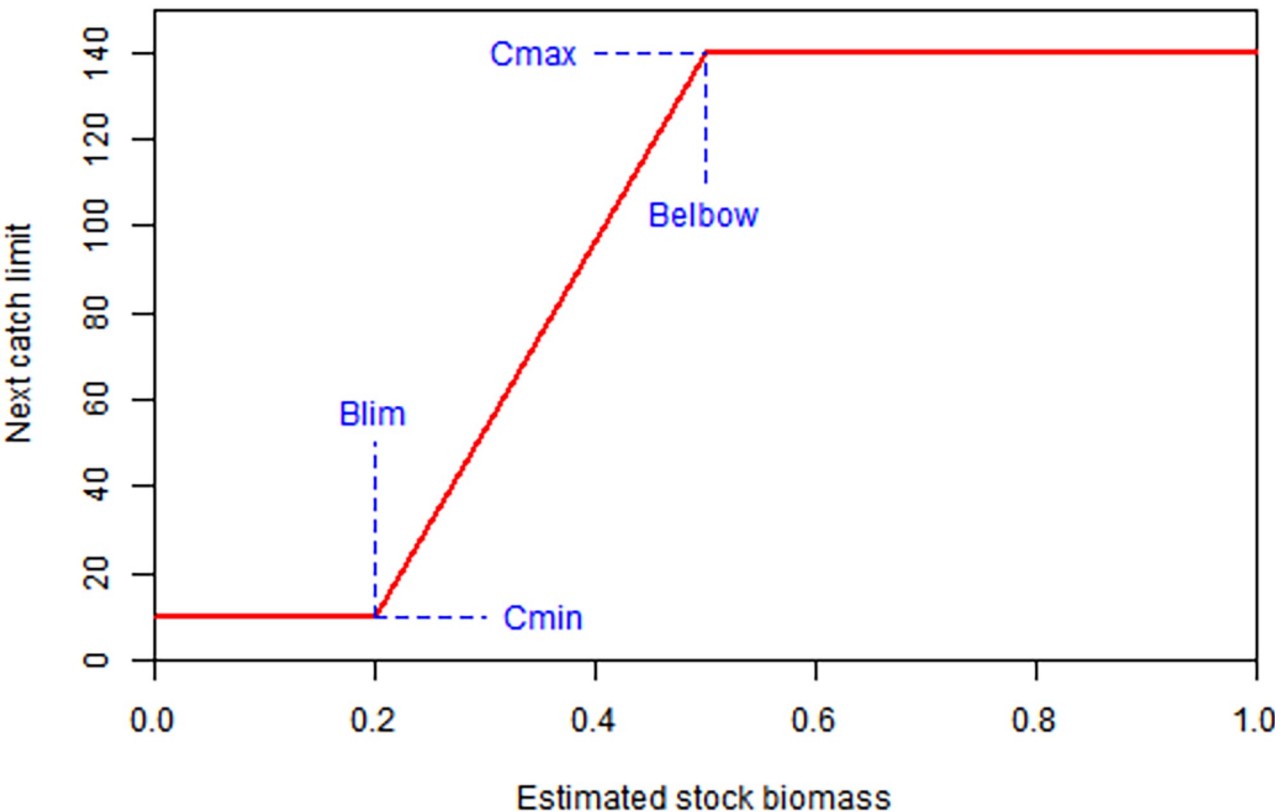

**Fig 1. The shape of the HCR.** The shape of the HCR is determined by four parameters: *Cmax*, *Cmin*, *Blim* and *Belbow* that provide the coordinates for the two inflexion points. In the example shown, *Cmax* = 140, *Cmin* = 10, *Blim* = 0.2 and *Belbow* = 0.5. Users can change each parameter to alter the shape of the HCR. The catch limit in the next year is based on the estimated stock biomass in the current year.

*Cmin* is generally set at some nominal level of catch greater than 0. *Belbow* does not necessarily represent a target biomass reference point because the long-term steady state expected value of biomass in the absence of uncertainty is given by a combination of all four parameter values. For example, an HCR with a long, shallow slope given by a high value of *Belbow* would likely result in steady state biomass part way up the slope, between *Blim* and *Belbow*. Instead, *Belbow* and *Cmax* should be chosen to provide the required performance of the HCR. This can be explored in the apps. In each of the three apps, users can change the parameters and modify the shape of the HCR.

A 'constant catch' option is also available as a comparison to using an HCR, i.e. the catch limit is fixed and does not respond to changes in the stock status.

In the real world, different sources of uncertainty can affect fisheries management and an adopted management procedure should be robust to this uncertainty [20, 21]. In AMPLE, uncertainty can be included in two ways: *biological variability* and *estimation error*.

Biological variability reflects the natural fluctuations in the stock dynamics, for example in recruitment, growth, and natural mortality. It is important that this source of uncertainty is considered when selecting an HCR. In AMPLE, biological variability is modelled by applying a noise term to the surplus production *g(B)*. The noise is positively autocorrelated, using the method in [22], so that there are periods of higher and lower than average growth.

Estimation error is the difference between the estimated and true value of stock status. As the true value of the stock status is never known, an HCR uses an estimated value to set fishing

opportunities, e.g. from a stock assessment model. This difference can strongly affect the expected performance of an HCR. There is no estimation method in AMPLE. Instead, an estimation method is simulated by applying normally distributed noise to the true value of biomass to generate an estimated value with error. It is also possible to introduce an estimation bias, so that the estimated biomass used by the HCR is consistently over or underestimated. Estimation bias is a real world problem and may mean that the HCR parameters *Blim* and *Belbow* need to be adjusted to compensate for it.

The standard deviations of the biological variability, the estimation error, and the level of estimation bias are set by users in the apps.

In AMPLE, the catch limit set by the HCR is always taken exactly, i.e. there is no implementation error.

The package includes vignettes for each app that act as self-guiding tutorials to facilitate learning outside of a workshop (see S1–S3 Files). They can also be used by instructors when running a training workshop. These tutorials are also accessible under the **Information** tab of each app, where a brief overview is also available.

In the package, it is assumed that the data collection and estimation method of the management procedure are fixed and the focus is on HCRs. Although the apps and accompanying vignettes refer to comparing HCRs, strictly speaking it is management procedures that are being compared.

AMPLE can be installed in R from CRAN and loaded in the usual way.

```
install.packages("AMPLE", dependencies = TRUE)
library(AMPLE)
```

The apps are also hosted online for users unfamiliar with R (see below).

## Performance indicators

The *Measuring performance* and *Comparing performance* apps include the calculation of several performance indicators that can be used to evaluate the performance of HCRs (Table 1).

See the main text for details on how these performance indicators are calculated. The indicators are presented as average values over three time periods (short-, medium- and long-term) and reported as the 90th percentile range and the median across the replicates.

**Table 1. Performance indicators calculated for the *measuring performance* and *comparing performance* apps.**

| Performance indicator | Description |
|---|---|
| Biomass | The 'true' biomass of the stock, relative to the unfished biomass (i.e. scaled between 0 and 1). This can be different to the estimated value of biomass used by the HCR when estimation error is present. |
| Probability that stock biomass > Limit Reference Point (LRP) | The proportion of replicates in each year in which the true relative biomass is greater than the LRP. It reflects the risk that the stock is overfished. It is presented as the probability of being above the LRP, rather than below, so that a larger value (maximum of 1) is better. |
| Catch | The expected catch of the fishery. |
| Relative catch-per-unit of fishing effort (CPUE) | The CPUE relative to the CPUE in the last historical year. |
| Relative effort | The fishing effort relative to the fishing effort in the last historical year. |
| Catch stability | Measures how much the catches change over time, scaled between 0 and 1. A value of 1 means that the catches are very stable and do not change at all. A low value means that the catches fluctuate strongly. |
| Proximity to the Target Reference Point (TRP) | Measures how close the relative biomass is to the TRP, scaled between 0 and 1. A value of 1 means that the biomass is always at the TRP. A low value means that the biomass is higher, or lower, than the TRP. |

The Limit Reference Points (LRP and TRP) are key biomass reference points for tuna stocks managed through the WCPFC. Both are expressed as relative to the unfished biomass. The LRP is a biomass limit, below which the stock is thought to be unable to sustain itself. In AMPLE the LRP is fixed at 0.2, in line with the WCPFC tuna stocks. The TRP is a biomass target which represents a 'sweet spot' in terms of the performance of the fishery and can include considerations such as the economic performance of the fishery and the distance from the LRP. Note that there is no economic analysis in AMPLE and the TRP is fixed at the arbitrary level of 0.5.

Annual fishing effort is calculated as:

$$E_y = C_y / \left( B_y \, q \right) \tag{2}$$

Where $q$ is the catchability of the stock by the fishing fleet. In AMPLE $q$ is fixed at 1.

Catch-per-unit of fishing effort (CPUE), sometimes known as the catch rate, is calculated as $CPUE_y = C_y / E_y$. As $q$ is currently fixed in time the CPUE is linearly related to biomass meaning that this indicator only provides limited additional information for HCR selection. However, this linear relationship may not hold for real world HCR evaluations and, given that CPUE is an important indicator of an HCR's performance, it is included for completeness. Additionally, future releases of AMPLE may include hyperstability, where $q$ is a function of biomass, so that CPUE is not linearly related to biomass.

Both effort and CPUE are reported as relative to the values in the last historical year. Reporting a relative value by comparing it to some value in the past can make it easier for users to interpret, for example by removing units. The last historical year (default 2019) is chosen as an arbitrary reference year. The duration of the historical period, and hence the last historical year, can be changed under the **Settings** tab.

Catch stability is calculated as a transformation of the absolute annual catch variability:

$$Cvar_y = \left| C_y - C_{y-1} \right| \tag{3}$$

$$Cstab_y = min\left( 1 - \frac{Cvar_y}{Cvar_{max}}, \, 0 \right) \tag{4}$$

Where $Cvar$ is the catch variability, $Cstab$ is the catch stability and $Cvar_{max}$ is the maximum expected catch variability, here set to a value of $K / 10$. This transformation gives a catch stability that ranges between 0 and 1, where 1 is equivalent to a catch variability of 0, i.e. the catches do not change at all, and 0 is the equivalent of high catch variability, set by $Cvar_{max}$.

A relative biomass close to the TRP is considered to be desirable. The proximity of the relative biomass to the TRP is calculated by scaling the absolute difference of the relative biomass and the TRP by the maximum possible distance from the TRP:

$$TRPprox_y = max\left( 1 - \frac{\left| B_y - TRP \right|}{max(TRP, 1 - TRP)}, \, 1 \right) \tag{5}$$

The proximity to the TRP has a value ranging between 0 and 1, where 0 is as far away from the TRP as possible, and 1 is exactly at the TRP. It does not distinguish between whether the relative biomass is higher or lower than the TRP.

The performance indicators are calculated as the average over three time periods (short-, medium-, and long-term). This is important as fisheries managers and stakeholders are not just interested in the long-term performance of the fishery.

For most indicators in Table 1, the higher the value the better (i.e. higher catches and higher catch stability are thought to be better than lower catches and catch levels that change a lot over time). However, higher fishing effort is not necessarily better as it may result in higher costs of fishing. Similarly, higher biomass might not be better as it may indicate foregone yield.

It is important that users fully understand the performance indicators so that they can be interpreted correctly. While using the apps, users are encouraged to make the link between the indicators and management objectives, i.e. if your priority was economic stability, which of these indicators would be the most important?

## Introduction to HCRs

The first app in the package introduces the basic ideas behind HCRs and how they can be used to manage fisheries.

To view the app vignette enter this command in R after loading the package:

```
vignette("intro_hcr", package = "AMPLE")
```

To launch the app enter:

```
intro_hcr()
```

When the app opens, the user will see time series plots of the catch and biomass in the main panel as well as a plot of the current HCR (Fig 2). The shape of the HCR can be changed using the controls in the left-hand panel of the app. A blue arrow connects the biomass plot to the HCR plot. This illustrates how the current estimate of biomass is used as an input to the HCR. As the biomass changes through time, the arrow changes shape and position to reflect the current estimate. The new catch limit, set by the HCR, is shown in the catch plot.

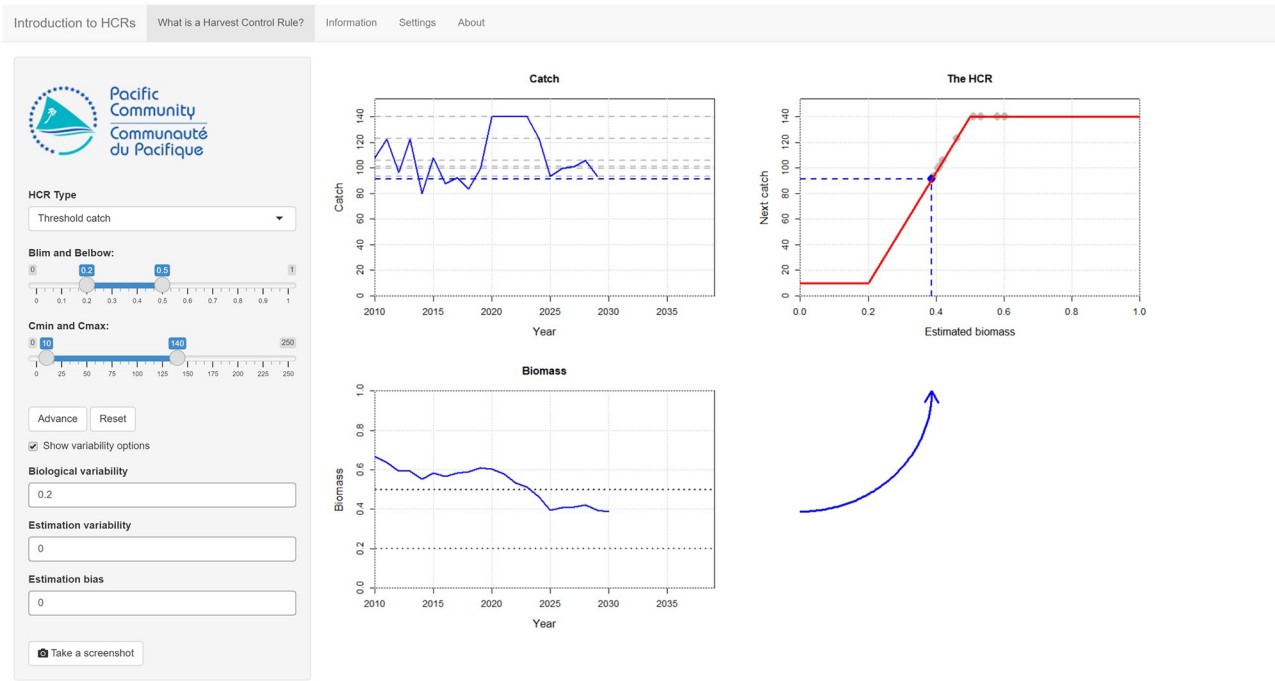

**Fig 2. A screenshot of the main page of the *Introduction to HCRs* app.** The controls that set the shape of the HCR and the variability options are in the left-hand panel. Biological variability has been switched on to include uncertainty in the result. In this example, the **Advance** button has already been pressed several times to project forward several years. The estimated stock biomass (the bottom left plot) is used by the HCR (the top right plot) to set the catch limit in the next year (the top left plot). The blue vertical dashed line on the HCR plot is the current estimated stock biomass. The blue arrow connects the biomass plot to the HCR. The shape and position of the arrow changes as the current estimated biomass changes to reflect how the input to the HCR changes. The catch limit in the next year is shown as the blue dashed horizontal line in the HCR and catch plots. Previous values are shown as grey dots on the HCR plot and grey dashed lines in the catch plot. The biomass plot shows the Limit Reference Point (LRP) and Target Reference Point (TRP) as dashed horizontal lines.

The app introduces the basic process of how an HCR can be used to set a catch limit based on the estimated stock status. Every time the **Advance** button is pressed, the fishery steps forward a single year. Users can see the catches reach the catch limit that was set by the HCR, the biomass respond to the stock being fished at the catch limit, and the updated estimate of biomass being used by the HCR to set the next catch limit. These steps move anti-clockwise around the app. By repeatedly pressing **Advance** users can follow the evolution of the fishery until the end of the full projection.

When the app is first opened the variability options are switched off and the projections are deterministic. After users become comfortable with HCRs, the different sources of uncertainty can be introduced by selecting the **Show variability options** box and setting appropriate values (the vignette suggests a biological variability of 0.2—there is nothing special about this value, it just gives a reasonable amount of variability).

This app has been very effective at explaining the basic mechanics of HCRs to workshop participants. By stepping through the process year-by-year, and seeing how the fishery evolves, users are able to gain a thorough understanding of how HCRs work in principle. The ability to experiment with different HCRs reinforces the idea that HCRs can perform differently and that selecting the 'right' HCR is important. Of key importance is the idea that rather than making a series of annual decisions on what the catch limit should be (as may be the case with traditional fisheries management), a single big decision is made on the shape of the HCR. The future fishery performance is then a consequence of this big decision.

## Measuring performance

When a management procedure, including the HCR, is evaluated with MSE many hundreds of projections with uncertainty, or replicates, are run where each replicate has a different realisation of the uncertainty. To fully understand the performance of the management procedure it is necessary to consider all the replicates together. Performance indicators are calculated for each replicate and summaries, such as the median and ranges of values, are presented to allow managers and stakeholders to decide which management procedure they prefer.

The *Measuring performance* app builds on the previous app and assumes that users are familiar with how HCRs work. It explores how the performance of a proposed HCR can be measured using performance indicators. Focus is put on the impact of uncertainty and the need to run many replicates to get a full understanding of the expected performance.

To view the app vignette enter this command in R after loading the package:

```
vignette("measuring_performance", package = "AMPLE")
```

To launch the app enter:

```
measuring_performance()
```

The front page of the app looks similar to the previous one so that users should already be familiar with the interface (Fig 3). There is an additional time series plot of the catch-per-unit of fishing effort (CPUE). To make the CPUE easier to interpret it is presented as relative to the CPUE in the last historical year (default 2019, noting that the length of the historical period can be adjusted in the **Settings** tab).

In the previous app, to run a full projection users had to click the **Advance** button many times. In this app when users click **Run projection** a full projection over all future years is run. The HCR is used during the projection to set the catch limit based on the latest estimate of stock biomass in each year.

Initially, there is no uncertainty in the projections so that each replicate for an HCR is the same. This is so that users can familiarise themselves with how the app works. However, the app is more useful when the projections are run with uncertainty switched on.

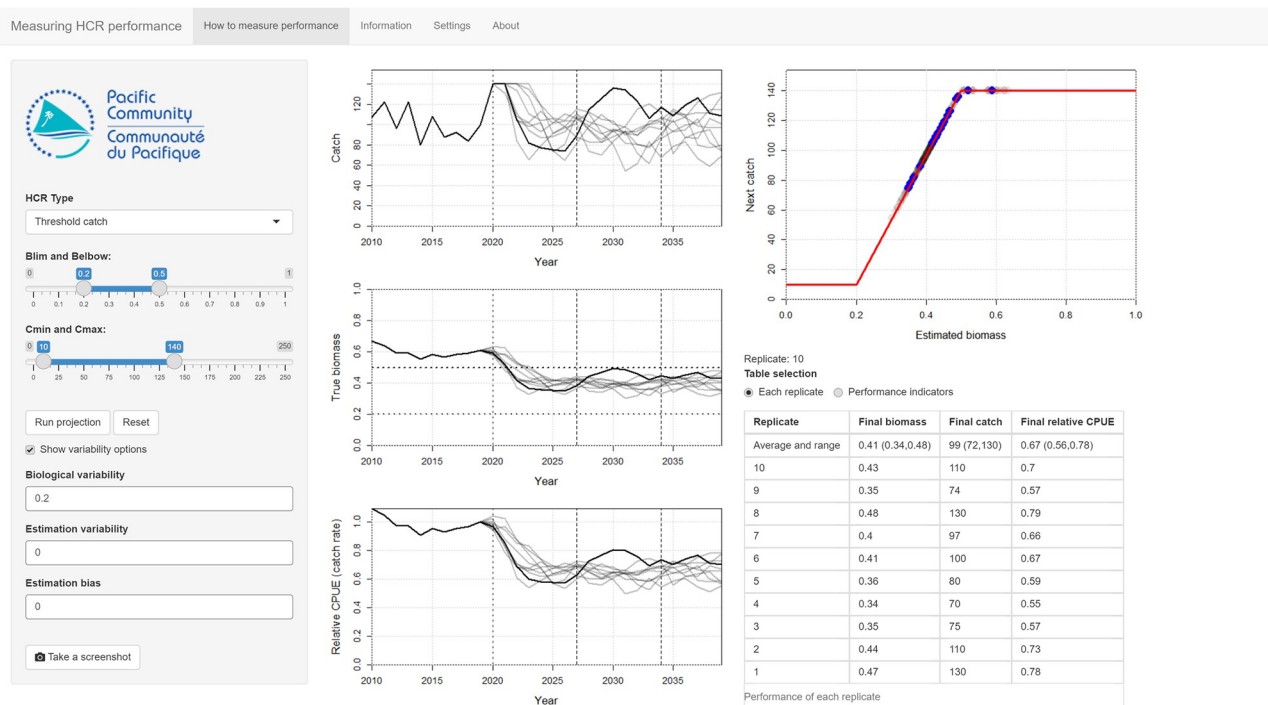

**Fig 3. A screenshot of the main page of the *Measuring performance* app.** The controls that set the shape of the HCR and the variability options are in the left-hand panel. Biological variability has been switched on to include uncertainty in the results. In this example, the **Run projection** button has been pressed ten times so that ten replicates have been run. The most recent replicate is shown as a black line in the time series plots. The previous replicates are shown as grey lines. On the HCR plot at the top right, the blue dots are the parts of the HCR that were 'active' in the most recent replicate. The grey dots are from the previous replicates. The table shows the biomass, catch and relative CPUE in the final year of the projection. The top row of the table gives a summary of these metrics as the median and the 90th percentile range.

Users are encouraged to press **Run projection** many times to run many replicates. The results of the current and previous replicates are shown in the plots. When 50 replicates have been run, the plots change to show a ribbon of the 90th percentile range, the median value and the last replicate.

The app has two ways of measuring performance, selected using the **Table selection** option under the HCR plot. The default option, **Each replicate**, is a table of results that reports the biomass, catch and relative CPUE in the final year of the projection for each replicate. The top row of the table reports summaries of the replicates as the 90th percentile range and the median value. As more replicates are run, users will see that the summaries begin to settle down. This is reinforced by the range of values seen in the plots. This table provides a simple way of measuring performance that familiarises users with the idea that many replicates must be run before the performance of an HCR is understood.

The second option, **Performance indicators**, is a table of indicators that capture a range of different information about the performance of the fishery (Table 1). The values in the table are the 90th percentile range (in parenthesis) across the replicates and the median.

## Comparing performance

Before a management procedure, including the HCR, is put into operation many candidate management procedures are tested with MSE. Their performance is compared using performance indicators and the preferred management procedure is selected on the basis that it is the one most likely to achieve the management objectives.

These concepts are explored in the *Comparing performance* app, in which users are able to test and compare the performances of multiple HCRs using performance indicators. As mentioned above, AMPLE uses the same data collection and estimation method for each HCR, so that comparing HCRs is effectively the same as comparing management procedures. By examining the indicators, and thinking about management objectives, a preferred HCR can be selected.

To view the app vignette enter this command in R after loading the package:

```
vignette("comparing_performance", package = "AMPLE")
```

To launch the app enter:

```
comparing_performance()
```

The front page of the app looks very similar to the *Measuring performance* app (Fig 4). Pressing the **Project** button runs 250 replicates of the same HCR. Biological variability is already turned on with a default value of 0.2. The results of the projection are shown in the time series plots and the performance indicators table and are the same as in the previous app. The number of replicates and the magnitude of the biological variability are adjustable in the **Settings** tab.

The basic process for testing and comparing HCRs is:

1. Select the shape of an HCR.

2. Press the **Project** button to run a projection and generate the plots and performance indicators.

3. Inspect the plots and the performance indicators in the table and, if acceptable, add the HCR to the basket by pressing **Add HCR to basket** (it is possible to name the HCR for clarity).

4. Go back to step 1 and try another HCR.

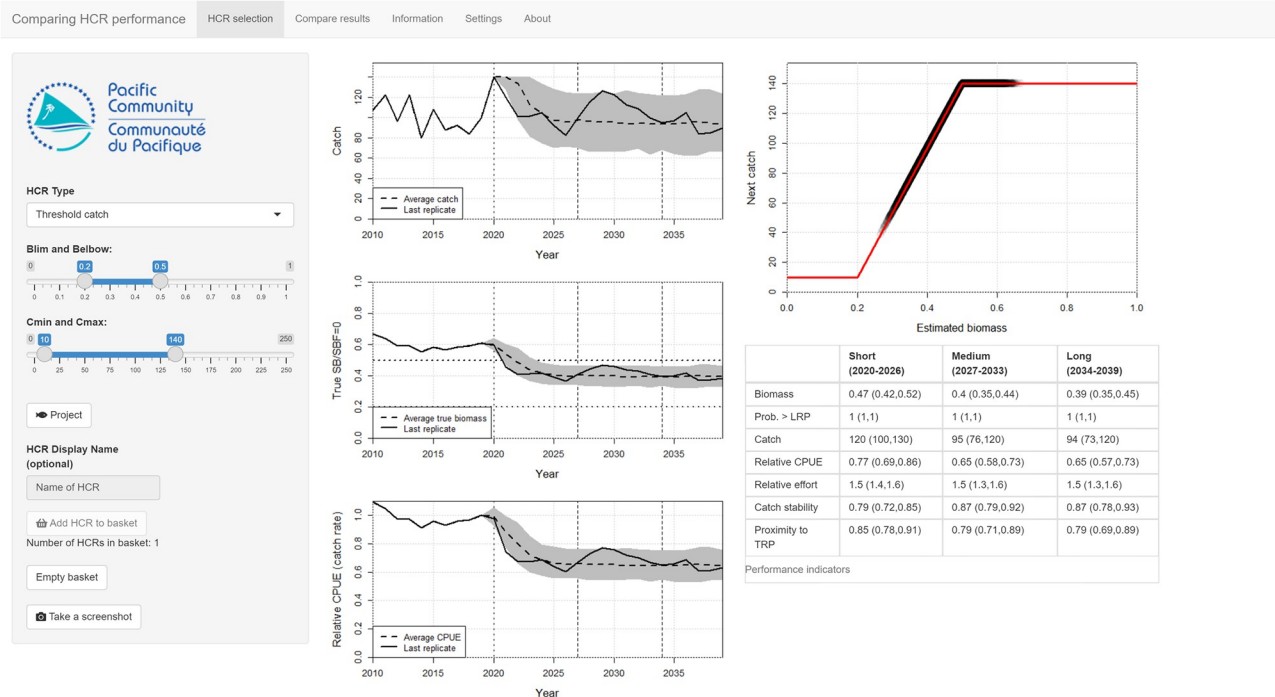

**Fig 4. A screenshot of the main page of the *Comparing performance* app.** The controls that set the shape of the HCR and run the projections are in the left-hand panel. In this example, the **Project** button has already been pressed to run a projection with multiple replicates. Biological variability is switched on by default. The results are shown in the time series plots. The grey ribbon shows the 90th percentile range, the dashed line is the median value across the replicates and the solid line shows the last replicate as an illustration. The performance indicators are shown in the table underneath the HCR plot. If users are happy with the results they can add the HCR to the basket of candidate HCRs by pressing the **Add HCR to basket** button.

When there are enough HCRs in the basket (at least three is recommended), their performance can be compared using the plots and tables in the **Compare results** tab.

The app includes three methods for comparing performance indicators across the candidate HCRs. The first is bar charts, under the **Performance indicators—bar charts** tab, where each panel shows the median value of an indicator for each HCR in each of the three time periods (Fig 5). Box plots are also available, selected by the **Performance indicators—box plots** tab, which report uncertainty in the indicator values and show the 80th and 90th percentile ranges, along with the median, in each time period. Finally, tables of results are found under the **Performance indicators—tables** tab, similar to that seen in the opening tab.

Only inspecting the median values of the performance indicators using bar charts does not give a full picture of the results as they do not consider uncertainty. However, not all users may be familiar with box plots, or similar types of plots that show a distribution of values. The bar charts are therefore included as a simplified way of exploring the results. Users are encouraged to use them as a 'stepping stone' towards a fuller consideration of uncertainty using the box plots.

There are three time periods and seven indicators with results reported with uncertainty for each HCR. As the number of candidate HCRs increases, the amount of information that users are required to process can become overwhelming. This is not dissimilar to what can happen when selecting a management procedure in the real world. To help with the selection process, it is possible to drop unwanted performance indicators and HCRs using the menus on the left-hand side of the app in the **Compare results** tab. Users are encouraged to consider which indicators best reflect their management objectives. The other indicators are therefore lower priority and can be dropped. Similarly, any candidate HCRs that do not meet the objectives can be

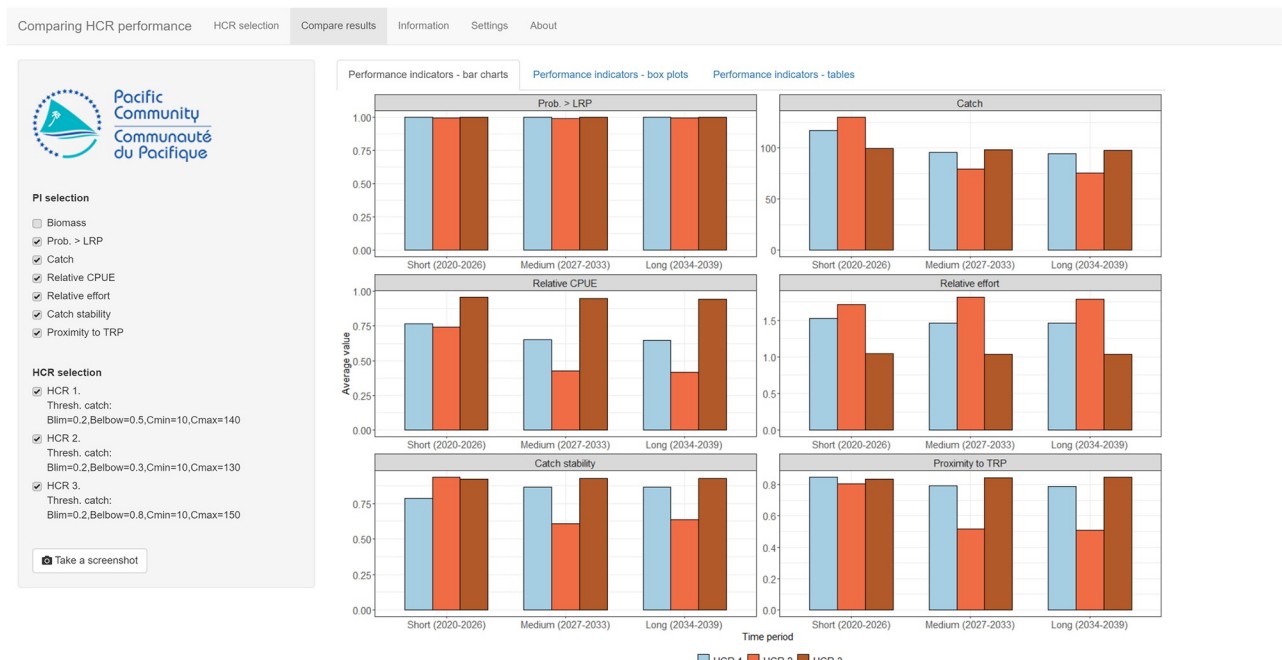

**Fig 5. A screenshot of the compare results tab in the *Comparing performance* app.** In this example, three HCRs have been tested and added to the basket of candidate HCRs. Each panel shows the median value of a performance indicator in each time period (short-, medium- and long-term) for each HCR. Here, the biomass indicator has been deselected leaving the remaining six indicators. Various trade-offs can be seen. For example, HCR 2 has the highest median catches in the short-term, but the lowest catches in the medium- and long-term.

dropped. By thinking carefully about management objectives, a process of elimination can be used to eventually select a preferred HCR.

During the WCPO training workshops, it was found that this app reinforced the message that the process of selecting a preferred HCR, or management procedure, in the real world is not a trivial task. Running group competitions and challenges led to lively discussions and it was interesting to see how the decision making process differed between groups. Getting all participants at a national workshop to agree on a single HCR reflected how challenging the process will be at a wider WCPFC level, given the number and diversity of WCPFC members.

## Future developments

There are several planned developments for the apps. Alternative HCR shapes will be included in future versions, including HCRs that set effort limits. Kobe and Majuro plots could be included as part of the comparison plots in the *Comparing performance* app.

The apps so far only consider a model-based type of management procedure, e.g. one that uses an estimate of biomass from a stock assessment. Empirical management procedures could also be included, e.g. by using the observed CPUE as an indicator of stock status.

## Conclusions

The AMPLE package provides three apps to support capacity building for harvest strategies. These apps were developed for training workshops around the WCPO and were found to be extremely useful in increasing attendee's knowledge of HCRs and supporting stakeholder participation in the WCPFC harvest strategy development process. They provide a thorough overview of HCRs and contain several novel features not found elsewhere. As the model fisheries in the apps are generic, they will also be of use to scientists and stakeholders developing harvest strategies in other regions.

## Supporting information

**S1 File. Introduction to HCRs tutorial.** The tutorial that accompanies the *Introduction to HCRs* app, available in the AMPLE package and through the online app.
(PDF)

**S2 File. Measuring performance tutorial.** The tutorial that accompanies the *Measuring Performance* app, available in the AMPLE package and through the online app.
(PDF)

**S3 File. Comparing performance tutorial.** The tutorial that accompanies the *Comparing Performance* app, available in the AMPLE package and through the online app.
(PDF)

## Acknowledgments

Thanks to Winston Chang for help with the R6 and Shiny reactivity interaction. A big thank you / *vinaka* / *ko rab'a* / *malo 'aupito* / *meitaki maata* to all the workshop participants who tried out various prototypes of these apps.

## Author Contributions

**Conceptualization:** Finlay Scott.

**Software:** Finlay Scott.

**Visualization:** Finlay Scott.

**Writing – original draft:** Finlay Scott, Nan Yao, Robert Dryden Scott.

**Writing – review & editing:** Finlay Scott, Nan Yao, Robert Dryden Scott.

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
