## [Decision Letter · Decision Letter 0]

9 Mar 2022

PONE-D-21-40367AMPLE: an R package for capacity building on fisheries harvest strategiesPLOS ONE

Dear Dr. Scott,

Thank you for submitting your manuscript to PLOS ONE. After careful consideration, we feel that it has merit but does not fully meet PLOS ONE’s publication criteria as it currently stands. Therefore, we invite you to submit a revised version of the manuscript that addresses the points raised by the reviewer, who is and expert on the field and has no competing interest. Please ensure that your decision is justified on PLOS ONE’s publication criteria and not, for example, on novelty or perceived impact.

We look forward to receiving your revised manuscript.

Kind regards,

Athanassios C. Tsikliras

Academic Editor

PLOS ONE

Journal Requirements:

Reviewers' comments:

Reviewer's Responses to Questions

**Comments to the Author**

1. Is the manuscript technically sound, and do the data support the conclusions?

Reviewer #1: Partly

2. Has the statistical analysis been performed appropriately and rigorously? 

Reviewer #1: I Don't Know

3. Have the authors made all data underlying the findings in their manuscript fully available?

Reviewer #1: Yes

4. Is the manuscript presented in an intelligible fashion and written in standard English?

Reviewer #1: Yes

5. Review Comments to the Author

Reviewer #1: This paper introduces an R package aiming to facilitate capacity building on Management Stratification Evaluation (MSE) and especially on a crucial part of the MSEs, on Harvest Control Rules (HCR). The package is not built for actual MSE design or HCR evaluation of performance, like DLMtools or the corresponding FLR packages, but rather as a tool for getting familiarized with the concept and the process of designing and evaluating HCR. As such, its potential use could be for training or teaching purposes. The package contains three interactive shiny apps, that takes you by the hand and walk you through the process of HCR designing, starting from the absolutely basics up to the comparison of different HCRs as a basis for decision making on management actions.

The app is well designed for its purpose. The structure of the app is intuitive and the navigation through the different tabs and pages is easy. More importantly, the app is very well documented; the developers/authors have prepared and provide users with well written tutorials, supporting every step of the learning process. Taking these into account, I believe that there is a place for the app in the MSE universe, despite the fact that alternatives, such as ToyTuna MSE tool, are available.

However, the paper introducing the app doesn’t meet the standards that the app itself has set. The paper in places seems to be a rather rushed summary of the tutorials and the underlying assumptions taken for model building are not well documented. For example, the authors introduce various metrics of HCR performance without adequately describing how these were estimated; they only provide a generic description of the relevant concepts. Probably these ‘details’ are not necessary for the tutorials. However, it is crucial to be included and clarified in the scientific paper supporting the app. Finally, an example-practical application of the app could benefit the paper a lot, but it is not provided/included by the authors.

Taking into account the above, my suggestion is for a major revision on the manuscript in order to be considered for publishing. See specific comments below.

Lines 83 -91

Although DLMtool is indeed not suitable for inexperienced users, since it is not built for capacity building but for actual MSE specification and testing, ToyTuna MSE tool is exactly that, a capacity building interactive tool, appropriate for users with no or limited knowledge on MSE and HCR and directly comparable with the app introduced by the users. Additionally, neither ToyTuna MSE tool, DLMtool nor AMPLE r package, after their release, could be modified by users to suit their needs. Finally, anyone is able to download DLMtool from CRAN repository and visit the ToyTuna MSE tool webpage without being restricted by any copyright issues. The authors should justify the novelty of their work on a better ground, especially in comparison with the ToyTuna MSE tool.

Lines 136-149

The authors should explain how this hockey-stick line is translated to a real-life harvest control rule-strategy and what the four introduced parameters represent. Additionally, Belbow is elsewhere referred as Btarget, which is more intuitive in my opinion.

Lines 241-243

The authors should explain how CPUE is estimated.

Lines 279-280 - Table 1

The authors introduce various performance indicators in the table without providing information on how these are estimated, other than a general description of what they represent. However, in order to be able to interpret the values of these metrics, as well as their relative importance, further information should be provided on the underlying assumptions as well as on how these metrics have been calculated. More specifically, the authors introduce ‘relative -to the last historical year- effort’. How the ‘last historical year’ effort has been assumed? How is the Limit Reference Point (LRP) set and how the Probability/Limit Reference Point (LRP) is estimated? The same applies also to the ‘Catch stability’ estimation. The authors define Target Reference Point (TRP) as “a 'sweet spot' in terms of economic performance of the fishery”, so is there an economical performance model running on the background and if yes, what is it? Are LRP and TRP connected with MSY? Even though the aim of the app is for learning purposes, the scientific paper supporting the app should clearly explain all the underlying assumptions for building it.

Lines 333-340

For comparing the performance of two or more HCRs the authors provide three tabs, one with bar charts, one with box plots and one with tables, all of which describe the same information. Although it is acceptable for an app like this to present the same information with a table and with some graphic representation, the fact that there are two tabs with two different chart types describing the same information could be confusing for some users. And in any way, I don’t think that a bar chart is appropriate for describing a median estimation, since a single value for an estimated variable without some measure of dispersion is usually meaningless in statistics. Any way, if authors insist on providing different graphic representations of their performance indicators (although box-plot do the job adequately) they should include a single tab for this task in their app, adding a button or a drop list through which the user would be able to change the form of the graphic representation. In this way, it would be clear for the user that they are working on the same set of results.

Lines 379

The link is broken. Replace the link with https://ofp-sam.shinyapps.io/AMPLE-comparing-performance/

Lines 397-399

The provided link is broken, replace with a valid one

Figure 1

Indicate Blim and Belbow on the x axis and Cmin, Cmax on the y axis, like figure 1 of S1.

6. PLOS authors have the option to publish the peer review history of their article (what does this mean?). If published, this will include your full peer review and any attached files.

Reviewer #1: No

---

## [Decision Letter · Decision Letter 1]

24 May 2022

AMPLE: an R package for capacity building on fisheries harvest strategies

PONE-D-21-40367R1

Dear Dr. Scott,

We’re pleased to inform you that your manuscript has been judged scientifically suitable for publication and will be formally accepted for publication once it meets all outstanding technical requirements.

Kind regards,

Athanassios C. Tsikliras

Academic Editor

PLOS ONE

Additional Editor Comments (optional):

Reviewers' comments:

Reviewer's Responses to Questions

**Comments to the Author**

1. If the authors have adequately addressed your comments raised in a previous round of review and you feel that this manuscript is now acceptable for publication, you may indicate that here to bypass the “Comments to the Author” section, enter your conflict of interest statement in the “Confidential to Editor” section, and submit your "Accept" recommendation.

Reviewer #1: All comments have been addressed

2. Is the manuscript technically sound, and do the data support the conclusions?

Reviewer #1: Yes

3. Has the statistical analysis been performed appropriately and rigorously? 

Reviewer #1: Yes

4. Have the authors made all data underlying the findings in their manuscript fully available?

Reviewer #1: Yes

5. Is the manuscript presented in an intelligible fashion and written in standard English?

Reviewer #1: Yes

6. Review Comments to the Author

Reviewer #1: All the issues raised have been addressed adequately by the authors. The manuscript should be accepted for publication.

7. PLOS authors have the option to publish the peer review history of their article (what does this mean?). If published, this will include your full peer review and any attached files.

Reviewer #1: No

---

## [Editor Report · Acceptance letter]

30 May 2022

PONE-D-21-40367R1 

AMPLE: an R package for capacity building on fisheries harvest strategies 

Dear Dr. Scott:

I'm pleased to inform you that your manuscript has been deemed suitable for publication in PLOS ONE. Congratulations! Your manuscript is now with our production department. 

Kind regards, 

on behalf of

Prof Athanassios C. Tsikliras 

Academic Editor

PLOS ONE